# Structure sensitivity of Cu and CuZn catalysts relevant to industrial methanol synthesis

Roy van den Berg[1], Gonzalo Prieto[1,†], Gerda Korpershoek[1], Lars I. van der Wal[1], Arnoldus J. van Bunningen[1], Susanne Lægsgaard-Jørgensen[2], Petra E. de Jongh[1] & Krijn P. de Jong[1]

For decades it has been debated whether the conversion of synthesis gas to methanol over copper catalysts is sensitive or insensitive to the structure of the copper surface. Here we have systematically investigated the effect of the copper particle size in the range where changes in surface structure occur, that is, below 10 nm, for catalysts with and without zinc promotor at industrially relevant conditions for methanol synthesis. Regardless of the presence or absence of a zinc promotor in the form of zinc oxide or zinc silicate, the surface-specific activity decreases significantly for copper particles smaller than 8 nm, thus revealing structure sensitivity. In view of recent theoretical studies we propose that the methanol synthesis reaction takes place at copper surface sites with a unique configuration of atoms such as step-edge sites, which smaller particles cannot accommodate.

[1] Inorganic Chemistry and Catalysis, Debye Institute for Nanomaterials Science, Utrecht University, Universiteitsweg 99, 3584 CG Utrecht, Netherlands. [2] Haldor Topsoe A/S, Haldor Topsøes Allé 1, DK-2800 Kongens Lyngby, Denmark. † Present address: Max-Planck-Institut für Kohlenforschung, Kaiser-Wilhelm-Platz 1, 45470 Mülheim an der Ruhr, Germany. Correspondence and requests for materials should be addressed to P.E.d.J. (email: p.e.dejongh@uu.nl) or to K.P.d.J. (email: k.p.dejong@uu.nl).

Supported metal catalysts play a pivotal role in the production of fuels and chemicals, in the purification of exhaust gases and in electrochemical energy conversion systems[1–6]. Surface science and computational chemistry have shown that the adsorption and catalytic properties of these materials depend on the surface structure of the metal[7–14]. According to current insights, the adsorption of reactants, intermediates, transition states and products at step-edge surface sites results in decreased energy barriers for dissociation reactions and for reactions involving a transition state with a specific geometry[10–13]. For $\sigma$-bond association reactions the energy barrier is often insensitive to the structure of the metal surface as intermediates and the transition state can be similarly stabilized on terrace sites or step-edge sites[10–13,15]. Nanosizing metal particles to a size below 10 nm typically alters the surface structure by changing the fraction of surface atoms that constitute different adsorption sites, while nanosizing metals with a significant $d$-valence electron density at the Fermi level to a size below 2 nm typically also alters the local density of states for a specific surface site[12,16–18]. In line with the current understanding, the effect of the metal particle size in the range between 2 and 10 nm on the activity and selectivity has been found to be strong for dissociation reactions or reactions involving the cleavage or formation of $\pi$-bonds, such as in methanation, Fischer–Tropsch synthesis and ammonia synthesis, but weak or absent for hydrogenation reactions that involve $\sigma$-bonds[5–13,15].

For decades, it has been debated whether the hydrogenation of CO and $CO_2$ to methanol by copper catalysts is sensitive or insensitive to the structure of the copper surface[19]. Several studies found a linear relationship between the exposed copper surface area determined with $N_2O$ refractive frontal chromatography and the activity, suggesting a structure-insensitive reaction[20–24]. On the basis of the measured copper surface areas reported in these studies, the particles were estimated to be larger than 10 nm. Recent studies have shown that $N_2O$ is not just oxidizing the copper surface but also oxidizing (partly) reduced zinc species on metallic copper or oxygen vacancies in the zinc oxide support[25–27]. The $N_2O$ refractive frontal chromatography method therefore depends on the structure of the investigated catalyst and the conditions before $N_2O$ exposure, and this might result in differences between the measured and the real exposed copper surface areas. In contrast to the structure insensitivity observed before, recent studies suggested that the methanol synthesis reaction might be sensitive to the copper surface structure[27–30]. Furthermore, it has been shown that the activity of copper also depends on the presence of zinc, be it in the form of zinc oxide or zinc silicate[22–24,28,29,31,32].

To settle this long-standing debate on the structure sensitivity of copper catalysts for the methanol synthesis reaction, we examined the catalytic performance of supported catalysts with different copper particle sizes and with or without a zinc component. More specifically, we have synthesized 42 catalysts with copper particle sizes in the range from 2 to 15 nm on Zn-free supports (silica and carbon), on supports containing zinc as Zn silicate or on supports incorporating Zn oxide. Furthermore, we have synthesized series of catalysts with similar copper particle sizes and various zinc loadings, to determine the minimum amount of zinc required to achieve optimum activity. As it has been shown that chemisorption techniques determining the copper surface area are sensitive to the catalyst structure and the preconditions, and in view of a presumably fast equilibration of the surface composition of especially the smallest particles (<8 nm) in response to changes in the thermodynamic potential of the gas phase, we have chosen to investigate the structure sensitivity of the methanol synthesis reaction by relying on X-ray diffraction and (scanning) transmission electron microscopy ((S)TEM) to determine the size and geometric surface area of the copper particles. The performance of all catalysts is investigated in the methanol synthesis reaction under industrially relevant reaction conditions of 260 °C and 40 bar, using a synthesis gas feedstock containing 23% CO, 7% $CO_2$, 60% $H_2$ and 10% Ar.

Catalysts containing zinc oxide are about 10 times more active and catalysts containing zinc silicate are about 5 times more active in the methanol synthesis reaction than catalysts without zinc. The differences in activity are ascribed to the promoting effect of zinc, which is affected by the thermodynamic stability of the zinc phase. The turnover frequency (TOF) decreases by a factor of about 3 going from 8 to 2 nm copper particles for both Cu and CuZn catalysts. Hereby, this work shows unequivocally that the methanol synthesis at industrially relevant conditions is sensitive to the copper surface structure. In view of recent theoretical studies, the observed effect of the copper particle size on the activity indicates that the methanol synthesis reaction predominantly takes place at surface sites with a unique configuration of several copper atoms such as step-edge sites, which smaller particles cannot accommodate.

## Results

**Copper particle size.** Cu(Zn)/$SiO_2$ catalysts were synthesized on different silica supports via a heat treatment of impregnated copper nitrate in $N_2$ or 2% NO/$N_2$ resulting in copper oxide and subsequent reduction in $H_2$ gas atmosphere resulting in metallic copper. The CuO and Cu particle sizes based on X-ray diffraction and (S)TEM measurements for the different samples were in the range from 2 to 15 nm. Representative TEM images and copper particle size distributions are shown in Fig. 1 while full details for all samples are collected in the Supplementary Information including the identification in the (S)TEM images of the copper particles in catalysts containing zinc (Supplementary Figs 1–18, Supplementary Tables 1–6 and Supplementary Methods). Similar crystallite sizes were found for copper oxide after heat treatment and for metallic copper after reduction and passivation, as the change in particle size on reduction is <15% based on the difference in density for the two phases (see Supplementary Table 1 for the actual Cu and CuO sizes as determined with X-ray diffraction and (S)TEM). Particle size distributions determined with (S)TEM after heat treatment were therefore representative for the size distribution of the copper particles after reduction. In the case of heat treatment in $N_2$ and reduction in $H_2$, the copper particles obtained a size of about 2–3 nm, irrespective of the support porosity and the presence or absence of zinc[31,33,34]. For these catalysts, the particle size measurements were based only on (S)TEM since the crystallite sizes were below the detection limit of X-ray diffraction. In the case of heat treatment in 2% NO/$N_2$ and subsequent reduction in $H_2$, the size of the copper particles, as determined by (S)TEM and X-ray diffraction, ranged from 3 to 15 nm and depended on the support porosity and copper loading[35,36].

Cu(Zn)/C catalysts were synthesized on high-surface area graphite (HSAG) or carbon xerogels with different pore sizes via a drying step and a reduction in $H_2$ (Supplementary Table 2). The size of the copper particles as determined with TEM was between 5 and 11 nm, and correlated to the average pore size of the support. For the series of catalysts on HSAG with varying Zn loading, the copper particle size as determined with TEM varied from 5–6 nm for the sample without zinc to about 9 nm for the sample impregnated with a solution containing 2 M copper nitrate and up to 2 M zinc nitrate.

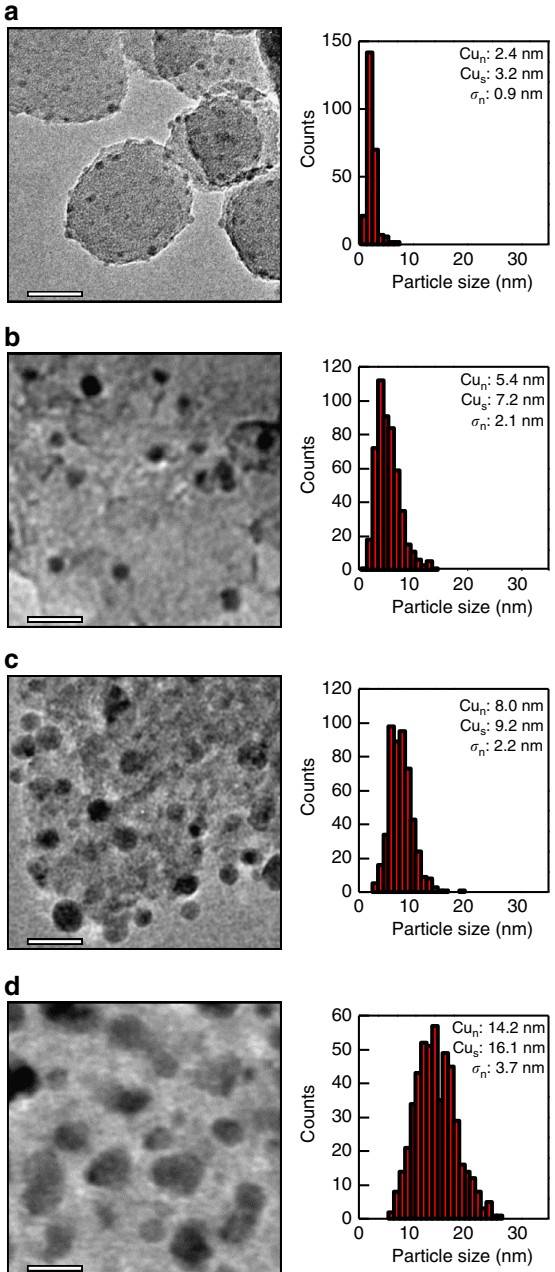

**Figure 1 | Electron microscopy images and corresponding particle size distributions of copper catalysts with particle sizes ranging from 2 to 15 nm.** The scale bars correspond to 20 nm. Number-average particle size ($Cu_n$), surface-average particle size ($Cu_s$) and s.d. of the number-averaged particle size ($\sigma_n$) are included in the figure. (**a**) Approximately 2.4 nm copper particles on silica after heat treatment in $N_2$ and reduction in $H_2$ ($Cu_2S$-$N_2$). (**b**) Approximately 5.4 nm copper particles on high-surface-area graphite after reduction ($Cu_9HSAG$). (**c**) Approximately 8.0 nm copper particles on silica prepared via homogeneous deposition precipitation followed by a hydrothermal treatment and reduction in $H_2$ ($Cu_{40}SiO_2$). (**d**) Approximately 14.2 nm copper particles on silica after heat treatment in 2% $NO/N_2$ and reduction in $H_2$ ($Cu_6SG(15)$-NO). The TEM images in **b** and **d** were acquired with a Tecnai 12 microscope and in **a** and **c** with an image-aberration corrected Titan microscope.

Cu/ZnO(/$Al_2O_3$) catalysts were synthesized via co-precipitation, followed by drying, calcination and reduction. The size of the resulting copper particles, as determined with TEM, increased

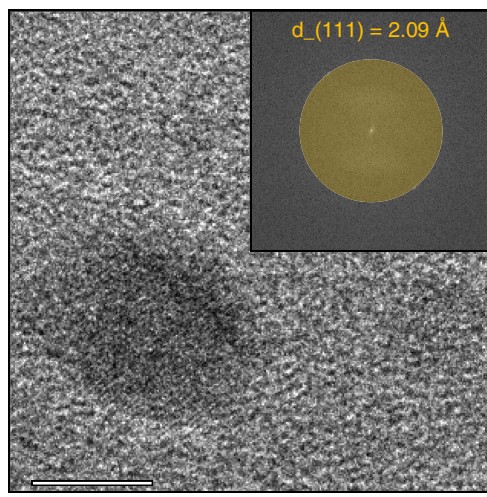

**Figure 2 | High-resolution *in situ* TEM image of $Cu_{40}SiO_2$ at 1 mbar $H_2$ at 250 °C.** The scale bar corresponds to 5 nm. The image on the top right shows the fast Fourier transform of the TEM image. The imaged copper particle is mono-crystalline. The distance between the observed lattice fringes is 2.09 Å, which corresponds to the (111) interplanar spacing of metallic copper crystals.

from 5–6 nm for the samples with a low copper loading (<10 wt%) to more than 30 nm for the samples with high copper loadings (>85 wt%, Supplementary Table 3). The Cu/ZnO samples with a copper loading close to that of commercial methanol synthesis catalysts (~55 wt%) had a copper particle size of about 12 nm. For the sample containing, in addition to copper and zinc, 11.6 wt% alumina, the copper particle size was ~7 nm, which is the typical size of the copper particles in commercial methanol synthesis catalysts[19,37]. Metallic copper crystallite sizes determined with X-ray diffraction were similar to the particle sizes determined with (S)TEM indicating that the particles were mono-crystalline. High-resolution TEM images (Fig. 2 and Supplementary Fig. 15) of copper catalysts under a reducing gas atmosphere showed mono-crystalline particles with the lattice spacing of metallic copper.

**Zinc species**. Depending on the catalyst formulation and the synthesis procedure, different zinc species can exist after reduction before catalysis. X-ray diffraction on CuZn/$SiO_2$ after reduction revealed the absence of crystalline ZnO in the sample (Fig. 3, red). This is in agreement with previous results from X-ray absorption spectroscopy that co-impregnation of copper and zinc led to the incorporation of zinc in the silica support resulting in zinc hydroxy(phyllo)silicate[31], which is more stable than ZnO and $SiO_2$ separately[38]. For the CuZn/$SiO_2$ samples after heat treatment in $N_2$ or 2% $NO/N_2$, zinc was thus most likely present in the form of zinc hydroxy(phyllo)silicate, which we refer to in this paper as zinc silicate. Similarly, X-ray diffraction of the CuZn/C samples after reduction did not show the presence of any crystalline zinc species (Fig. 3, blue). However, after 10 days of methanol synthesis, X-ray diffractions corresponding to ZnO were detected for samples with a Zn/(Cu + Zn) atomic ratio higher than 0.1 (Fig. 3, green). Hence, for carbon-supported catalysts, the Zn promoter was present as highly dispersed, amorphous ZnO before catalysis, which crystallized after prolonged exposure to reaction conditions. X-ray diffraction of Cu/ZnO(/$Al_2O_3$) samples after reduction showed the presence of crystalline ZnO with crystallite sizes around 12 nm (for Cu/ZnO see Fig. 3, black).

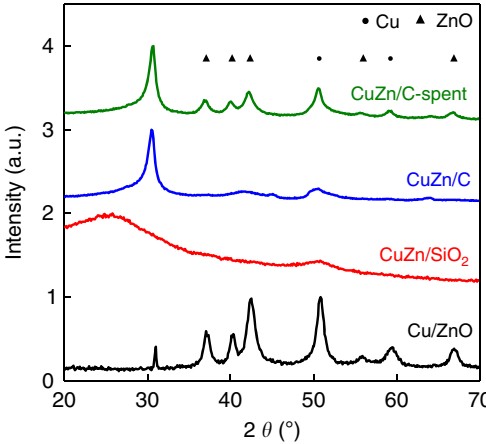

**Figure 3 | X-ray diffractograms of different types of zinc-containing catalysts.** X-ray diffractograms of Cu/ZnO ($Cu_{54}Zn$ after reduction, black), $CuZn/SiO_2$ (Cu$_8$ZnSG(11)-NO after reduction, red) and CuZn/C before ($Cu_8Zn_{8.0}$HSAG after reduction, blue) and after methanol synthesis ($Cu_8Zn_{8.0}$HSAG after catalysis, green) are shown. Diffraction peak positions relating to Cu and ZnO are marked with filled circles and triangles, respectively. The broad peak at $2\theta = 25°$ for the $CuZn/SiO_2$ sample is due to the silica support and the peaks at 31° for the CuZn/C and CuZnO samples are due to the carbon support and the graphite lubricant used for pelletization, respectively.

**Element distribution**. To determine the spatial distribution of copper and zinc in the different types of catalysts, STEM-energy dispersive X-ray spectroscopy (STEM-EDX) was used. In the case of $CuZn/SiO_2$ after heat treatment in 2% $NO/N_2$ and reduction in $H_2$, zinc was located throughout the support with an increased loading near high-density domains of copper particles (Supplementary Fig. 9)[31]. In the case of reduced and passivated CuZn/C, both copper nanoparticles and zinc species were found in close intimacy, showing local Cu/Zn ratios close to the bulk Cu/Zn ratios (Supplementary Figs 10 and 11). For reduced and passivated $Cu/ZnO(/Al_2O_3)$ samples, STEM-EDX showed that particles with a high copper content were located next to particles with a high zinc content (Supplementary Figs 12–14). For all three different types of catalysts, zinc was thus detected in close proximity to the copper particles.

**Effect of zinc on activity**. The activity and selectivity in the methanol synthesis reaction of the different catalysts were investigated at 40 bar (23% CO, 7% $CO_2$, 60% $H_2$ and 10% Ar), 260 °C and $CO + CO_2$ conversions below 20%, that is, below the thermodynamic conversion limit of 28.5%. The selectivity towards methanol was above 98.5% for all catalysts. The methanol synthesis productivity was calculated from the conversion of $CO + CO_2$ (Supplementary Methods). TOFs were calculated per surface metal atom and the dispersion was based on (S)TEM particle size distributions assuming fully accessible spherical copper particles. No correlation between silica support pore size and TOF was observed for samples with similar copper particle sizes, indicating that the copper particles were similarly accessible for methanol synthesis irrespective of the support porosity (Supplementary Fig. 18).

Copper catalysts without Zn and a copper particle size larger than 8 nm had a maximum TOF after 2–10 h on stream of about $2.5 \bullet 10^{-3} s^{-1}$, in close accordance with previously reported values for copper on silica under similar conditions (Supplementary Table 1; see Supplementary Fig. 16 for typical activity plots as a function of time on stream)[21–23]. Catalysts

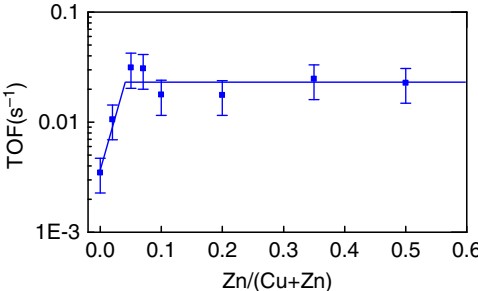

**Figure 4 | Effect of zinc loading on activity.** The surface-specific activity (TOF) for methanol synthesis after 2–10 h on stream at 260 °C and 40 bar plotted as a function of zinc loading for CuZn/HSAG catalysts. Error bars indicate the s.d. in the methanol synthesis activity (35%).

containing Cu particles and zinc hydroxy(phyllo)silicate (Cu/Zn-silicate) were about 5 times more active while catalysts containing Cu/ZnO were about 10 times more active than Cu catalysts (Supplementary Tables 2 and 3). The TOF for copper particles larger than 8 nm promoted by ZnO was about $2.5 \bullet 10^{-2} s^{-1}$, in reasonable agreement to values previously reported in literature[21,22,28,39].

The catalytic performance of the CuZn/HSAG catalysts with similar copper particle sizes but varying amounts of ZnO is shown in Fig. 4. The activity increased on increasing zinc loading until the maximum activity was reached at a $Zn/(Cu + Zn)$ atomic ratio of 0.05. A further increase in zinc loading did not change the activity significantly. For Cu/ZnO samples the amount of zinc oxide required to obtain maximum site-specific activity corresponds to a $Zn/(Cu + Zn)$ atomic ratio of about 0.1 (Supplementary Fig. 17), similar to previous reports[40,41]. Since the zinc-containing catalysts used to study the effect of the copper particle size all have a zinc loading substantially higher and STEM-EDX showed for the different types of catalysts that zinc was located in close proximity to the copper particles, it is concluded that the promotion by zinc was not limited by the zinc loading for those catalysts. The observed promotion effect for zinc-containing catalysts thus seems to be inherent to the nature and related thermodynamic stability of the zinc phase, that is, the thermodynamically less stable ZnO has a stronger promoting effect than the more stable zinc silicate[38].

Many mechanistic models have been proposed in literature to explain the role of zinc, including hydrogen spillover from zinc oxide to copper[42], the stabilization of either $Cu^+$ or $Cu^-$ species[43,44], shape changes of the Cu nanoparticles on zinc oxide supports[45] and zinc-induced defects in the copper structure[28,40]. Recent work indicates that the increase in TOF for zinc-containing catalysts is mainly due to the formation of (partially) reduced zinc species from zinc oxide or zinc hydroxy(phyllo)silicate and subsequent migration to the copper surface, increasing the copper surface-specific activity[26,28,29,46–49]. Since this process is affected by the thermodynamic stability of the zinc phases involved, we conclude that for the catalysts containing Zn-silicate the equilibrium amount of zinc species migrated on the surface of the Cu nanoparticles is lower than for samples containing ZnO, which led to a stronger zinc promotion effect for the latter samples.

**Effect of copper particle size on activity**. For all three types of catalysts the TOF decreased for copper particle sizes below about 8 nm (Fig. 5). Within the uncertainty, a decrease of the TOF by a factor of about 3 is apparent going from particles of 8 to 2 nm for catalysts containing Cu, Cu/Zn-silicate and Cu/ZnO although the data for the latter do not allow a quantitative conclusion

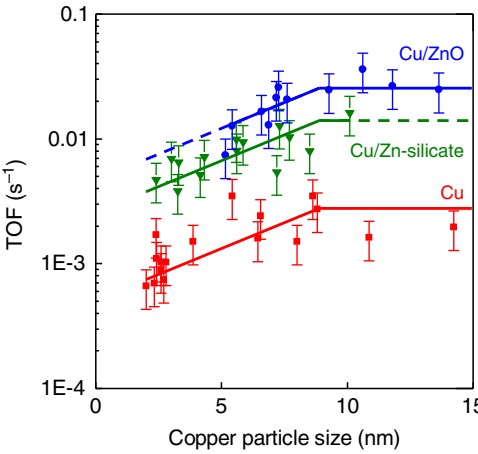

**Figure 5 | Effect of copper particle size on activity.** The surface-specific activity (TOF) for methanol synthesis after 2–10 h on stream at 260 °C and 40 bar plotted as a function of the number-averaged copper particle size for catalysts containing copper, that is, Cu/SiO$_2$, Cu/(functionalized)SiO$_2$ and Cu/C (red), containing Cu/Zn-silicate, that is, CuZn/SiO$_2$ (green), and containing Cu/ZnO, that is, Cu/ZnO(/Al$_2$O$_3$) and CuZn/C (blue). Error bars indicate the s.d. in the methanol synthesis activity (35%). For the s.d. in the particle size, see Supplementary Figs 5–7. Trendlines were added to guide the eye.

concerning the effect of the particle size below 5 nm. To obtain highly active copper-based methanol synthesis catalysts it is thus most important to promote catalysts with zinc as it can increase the activity by a factor of 10, but it is also important to tune the copper particle size as it can further increase the activity by a factor of 3.

Neither for non-reducible oxide supports such as SiO$_2$ nor for carbon supports a strong interaction between metal particles larger than 2 nm and the support is expected[50]. This is confirmed by the large contact angle of 135° that has experimentally been found for copper particles on silica in reducing conditions[34]. Such a large contact angle allows approximating the copper surface area by assuming fully accessible spherical particles. Furthermore, it has been reported that the energy of the copper–silica interface does not depend on the gas environment, and that the contact angle of copper particles larger than 2 nm on silica does not depend on particle size[34,45]. The observed particle size effect for the copper catalysts without zinc can thus not be due to support effects but rather seems to be due to changes in the structure of the copper surface.

For copper on zinc-containing supports, strong metal–support interactions have been reported in literature. It has been observed that for copper on ZnO both the shape of the copper particles, as well as the coverage of the copper particles with (partly) reduced zinc species depend on the gas environment[27,45]. The coverage with zinc species might depend on the copper particle size, and could result in a decreased activity for smaller copper particles[27]. Although we cannot fully exclude that support effects play a role in the observed particle size effect for zinc-containing supports, the similarity of the observed particle size effect for catalysts with and without zinc strongly suggests that the lower activity for small particles is mainly due to changes in the copper surface structure. Hence, decorating the copper surface with migrated (partially) reduced zinc species increases the activity up to a factor of 10, but does not seem to significantly alter the configuration of the active site[28].

The TOF per surface atom depends on the ratio between active sites and surface atoms, the activation energy of the rate-

determining step and the active site coverages of the reactants in that step. Nanosized metallic particles have surface structures with a size-dependent fraction of different surface sites[16,17]. Every type of surface site provides an atomic arrangement with its own bond-making and bond-breaking capabilities resulting in a specific activation energy and reactant coverage. For particles larger than ca. 2–3 nm and with a significant $d$-valence electron density at the Fermi level, the local density of states for a specific surface site is considered to be size-insensitive[12,18]. Particle size effects for particle sizes larger than 2–3 nm are therefore mostly due to changes in the fractions of sites with different activities.

The fraction of active sites normalized per surface atom depends on the specific atomic configuration of the active site and on the geometry of the metal surface. Reactants, intermediates, transition states and products of catalytic reactions often show stronger adsorption on more unsaturated surface sites[7–14]. The transition states of reactions that involve the cleavage or formation of molecular $\pi$-bonds are in general additionally stabilized by interaction with active sites consisting of multiple atoms, thereby lowering the energy difference between the reactants and the transition state for both the forward and backward reactions[10–13]. For example, the cobalt-catalysed Fischer–Tropsch reaction and ruthenium or iron-catalysed ammonia synthesis have been proposed to require a reaction centre with a unique configuration of several metal atoms, which smaller particles cannot accommodate, for example, B5 and C7 sites[6–13,17,51,52]. Reactions involving dissociation of $\sigma$-bonds are often also structure-sensitive as the transition state adsorbs stronger on more unsaturated surface sites. The transition states of reactions that involve the cleavage or formation of $\sigma$-bonds do, however, typically not require a unique configuration of several metal atoms[10–13]. As intermediates and transition states for formation of $\sigma$-bonds are often similarly stabilized on close-packed and on more unsaturated sites, the energy barriers for these reactions are normally relatively insensitive to the structure of the metal surface[10–14]. For example, the structure sensitivity is typically weak or absent for many hydrogenation reactions[7–15].

Recently, in the case of the methanol synthesis reaction, density functional theory (DFT) calculations were reported for a higher Miller-index copper surface, Cu(211), which contains step sites, such as B5 sites, and copper step atoms with lower coordination numbers compared with atoms at the most densely packed copper surface, Cu(111). The DFT calculations predict that the energies of both the intermediates, as well as the transition states are generally lower at the Cu(211) than at the Cu(111) surface[28]. As a result, the adsorption of (intermediate) surface species at Cu(211) is stronger and the energy barriers for the elemental steps in the formation of methanol different due to species-specific changes in adsorption strength. Overall, the energy difference between reactants and least stable transition state decreased, indicating that the Cu(211) surface is more active. At the same time, the stronger adsorption of formate at the higher Miller-index facet results in a high formate surface coverage for CO$_2$ containing synthesis gases. High formate coverages can have a poisoning effect on the hydrogenation of CO and at high coverages also on the hydrogenation of CO$_2$ due to the unavailability of the metal surface sites to other reaction intermediates[39,53]. Isotopic labelling studies show that CO$_2$ is the main source of methanol in synthesis gases containing both CO and CO$_2$ (ref. 39).

Considering these aspects, the observed decrease in activity for particles smaller than 8 nm can be rationalized along two lines. The first possibility is that the active site consists of a unique configuration of copper atoms, in line with the suggestion by Behrens *et al.* that step sites are the active sites, since for a Cu(211) surface the overall energy barrier was calculated to be

lower[28]. Van Helden *et al.* have computed for cobalt face-centered cubic particles that the site fraction of step sites increases with increasing particle size[17]. The fraction of so-called B5A and B6 step sites stabilizes above 4 nm and the fraction of B5B step sites above 8 nm. Although the equilibrium shapes under reaction conditions for copper particles might be slightly different, these calculations show that the observed particle size effect up to 8 nm might be due to changes in the fraction of specific step sites.

Another possibility is that the activity of more unsaturated surface sites is inhibited by formate and that the reaction predominantly takes place on sites at more densely packed copper surfaces, which are not poisoned by formate. In this case the copper particle size effect is due to the increase in the fraction of atoms with lower coordination numbers below 8 nm (ref. 16). A microkinetic evaluation of Janse van Rensburg *et al.* using DFT energies for Cu(111) and Cu(211), suggests that for more unsaturated surface sites, the lower overall energy barrier dominates and that changes in the fraction of specific step sites contribute most to the observed particle size effect[53].

## Methods

**Synthesis.** Cu(Zn)/SiO$_2$ catalysts were synthesized on different silica supports (silica gels, (aminopropyl-functionalized) Stöber silica, SBA15, SBA16 and FDU12) via a heat treatment of impregnated copper nitrate in N$_2$ or 2% NO/N$_2$, and subsequent a reduction in H$_2$. Cu/SiO$_2$ was also prepared via homogeneous deposition precipitation of copper on colloidal silica followed by a hydrothermal treatment and reduction. Cu(Zn)/C were synthesized on HSAG and carbon xerogels with different pore sizes via impregnation followed by a drying step and a reduction in H$_2$. Cu/ZnO(/Al$_2$O$_3$) catalysts were synthesized via co-precipitation, followed by drying, calcination and reduction.

**Characterization.** X-ray diffraction was performed with a Bruker-AXS D2 Phaser powder X-ray diffractometer equipped with a Lynxeye detector, in Bragg–Brentano mode with a $\theta$-$\theta$ system with a 141 mm radius. The radiation used is Co-K$\alpha$12 ($\lambda = 1.79026$ Å), operated at 30 kV, 10 mA.

TEM images were acquired with a Tecnai 12 (FEI) microscope operated at 120 kV or an image-aberration corrected Titan 80–300 ETEM (FEI) operated at 300 kV. High-angle annular dark-field STEM (HAADF-STEM) and EDX spectroscopy was performed using either a Tecnai 20FEG (FEI) electron microscope equipped with a field emission gun, a Fischione HAADF detector and an EDAX Super Ultra Thin Window EDX detector, or a TALOS F200x (FEI) electron microscope equipped with a field emission gun (XFEG), a Fischione HAADF detector and a SuperX EDX system. For these measurements, samples were placed onto a carbon-coated Ni TEM grid (Agar 162 200 Mesh Ni) and mounted either on a low-background sample holder (Philips) with a 0.1 mm-thick Be specimen support film and a Be ring to clamp the grid, and inserted in the Tecnai 20FEG or on a high-visibility low-background double-tilt sample holder (FEI) with a Be clamp, and inserted in the TALOS F200x. The signal and resolution in the different electron microscope systems was in general sufficient to detect copper particles larger than 2 nm on any of the supports. For each sample, at least 150 particles were analysed with respect to their size by measuring the diameter of a spherical approximation of the projected particle area.

**Catalytic testing.** The performance of the catalysts in the methanol synthesis reaction was investigated in a fixed-bed stainless-steel reactor with an inner diameter of 0.9 cm (Autoclave Engineers). The calcined Cu(Zn)/SiO$_2$ and Cu/ZnO(/Al$_2$O$_3$) samples, and the reduced and passivated Cu(Zn)/C samples were pressed, ground and sieved to obtain a granulate size of 0.42–0.63 mm. A unit of 0.03–0.9 g catalyst was diluted with 0–3 ml SiC granules (sieve fraction of granulate size 0.25–0.42 mm) and loaded into the reactor (catalyst bed height of 3–7 cm). Subsequently, the samples were (re)reduced *in situ* at 250 °C (2 °C min$^{-1}$) for 2.5 h with a flow of 110 ml min$^{-1}$ 20% H$_2$/Ar. After that, the temperature was lowered to 100 °C to prevent premature production of methanol when switching to synthesis gas. The reactor was flushed with synthesis gas (10% Ar, 7% CO$_2$, 23% CO and 60% H$_2$, Linde), which was purified with a metal carbonyl trap (4.0 g of 0.5–1.5 mm H-USY zeolite, CVB-780 from Zeolyst Int., and 5 g activated carbon, Norit R3B). The argon in the synthesis gas feed acted as an internal standard for the gas chromatograph (GC). After 30 min of flushing the pressure was increased to 40 bars. Subsequently, the exit gas composition was analysed every 110 min with a GC (Varian 450). The lines from the reactor to the GC were heated to 150 °C to avoid any methanol or water condensation. The first GC channel consisted of a HAYESEP Q (0.5 m × 1/8 in) column followed by a MOLSIEVE 13x (15 m × 1/8 in) column that led to a thermal conductivity detector. The second GC channel consisted of a CP-SIL 8CB FS capillary column that led to a flame ionization detector. At least three chromatograms were recorded of the synthesis gas feed. The temperature was then

increased to 260 °C (2 °C min$^{-1}$) to initiate methanol production. The amount of catalyst and the synthesis gas flow (10–30 ml min$^{-1}$) were chosen such that CO + CO$_2$ conversion levels below 20% were obtained. The initial (maximum) activity of the catalysts was determined by the maximum conversion of CO and CO$_2$ after about 2 to 10 h on stream. The s.d. in between different tests in the activity is ± 35%. TOFs were calculated per surface metal atom. The dispersion was based on (S)TEM particle size distributions assuming fully accessible spherical copper particles and calculated with the formula: dispersion $= \frac{6V_m}{A_m \times PS} = \frac{1.04}{PS}$, where $V_m$ is the molar volume of copper, $A_m$ the molar area of copper and PS the surface-average particle size in nanometre. The selectivity was determined from the flame ionization detector chromatograms and was in all cases more than 98.5% towards methanol, with trace amounts of dimethyl ether, methane, ethanol, ethane and propane. After catalysis, the samples were passivated for 15 min by exposing the sample to air diluted with Ar at room temperature. The samples were stored in a glove box under argon atmosphere.

Additional details about synthesis, characterization and catalytic testing can be found in the Supplementary Information.

**Data availability.** All data are available from the authors on reasonable request.

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

## Acknowledgements

The project was supported financially by Haldor Topsoe A/S. K.P.d.J. acknowledges the European Research Council, EU FP7 ERC Advanced Grant no. 338846. P.E.d.J. acknowledges The Netherlands Organization for Scientific Research NWO-Vici programme. Dr Jens Sehested is acknowledged for his suggestion to explore CuZn catalysts with different thermodynamic stabilities of the Zn precursors. Dr Stig Helveg is thanked for his contribution to the electron microscopy studies. Sebastian Kuld is acknowledged for TPR-MS measurements, Hans Meeldijk for STEM-HAADF and EDX measurements, Ceri Zwart, Elian Griffioen, Jochem Blum and Luc Smulders for the synthesis of CuZnHSAG samples, and Tanja Parmentier for the synthesis of Cu on aminopropyl-functionalized silica. We are grateful to Robert Goedvree of Grace for providing the silica gel supports.

## Author contributions

R.v.d.B. synthesized, characterized and tested catalysts, and drafted the manuscript; G.P. synthesized, characterized and tested CuZn/SiO₂ catalysts; G.K., L.I.v.d.W., A.J.v.B. and S.L.-J. synthesized Cu/SiO₂, Cu/C, CuZn/C and Cu/ZnO(/Al₂O₃) catalysts, respectively; P.E.d.J. and K.P.d.J. contributed to experimental design, data analysis and manuscript writing.

## Additional information

**Competing financial interests:** The authors declare no competing financial interests.

**How to cite this article:** van den Berg, R. *et al.* Structure sensitivity of Cu and CuZn catalysts relevant to industrial methanol synthesis. *Nat. Commun.* **7,** 13057 doi: 10.1038/ncomms13057 (2016).

