## [Peer review file · Nature Communications]

Reviewers' comments:

Reviewer #1 (Remarks to the Author):

This is an interesting and very clearly written manuscript in which the authors endeavor to prove that the conversion of synthesis gas to methanol over copper catalysts is sensitive to the structure of the copper surface. An exhaustive study has been made of Cu particles of various sizes dispersed on zinc-free supports, and on zinc silicate and zinc oxide supports.

The main conclusions of the paper are:-

- (i) Zn acts as a promoter, and its efficacy depends on the thermodynamic stability of the zinc phase, with less stable zinc containing phases giving rise to bigger promotion effects.
- (ii) the turnover frequency for Cu containing particles decreases by a factor of ~ 3 on decreasing their mean size from 8 to 2nm, irrespective of the support identity.

Both of these trends are clearly demonstrated by the experimental results and subsequent analysis presented. In addition, the experimental techniques used are appropriate for the task in hand and are well executed. In fact, the meticulous and detailed presentation of the methods and results in the SI is to be commended. It would have been nice to include some higher (atomic) resolution imaging studies of the supported particles to confirm they were indeed Cu rather than CuOx in the STEM, and to investigate more definitively if their exposed surface facet planes, defect (e.g. twin) content, and substrate wetting characteristics were varying with particle size and support identity.

Based on their experimental measurements of the dependence of turnover frequency on metal particle size, the authors postulate that "the methanol synthesis reaction predominantly takes place at surface sites with a unique configuration of several copper atoms such as step-edge sites, which smaller particles cannot accommodate." It would strengthen further the paper if some modelling studies more specific to this materials system and also incorporating the possible effects of Zn decoration of the Cu could be performed to complement this important postulate.

The paper relates to an industrially important process which with a little more work could potentially be of interest to the readers of Nature Communications.

Reviewer #2 (Remarks to the Author):

The paper by van den Berg et al. is a thorough and conclusive study of the structure and composition sensitivity of Cu(Zn) catalysts in the methanol synthesis reaction.

The authors present a large series of skillfully prepared and well characterized catalysts and report a clear trend of the TOF with absence or presence of the Zn promoter and - importantly - a breakdown in TOF for small Cu crystallites, similar to effects observed previously e.g. in ammonia synthesis over Ru.

Neither the effect of the Zn promoter nor the structure sensitivity by themselves are new, but the effect of very small particles is to my knowledge unprecedented for MeOH synthesis. I was impressed by the careful and detailed microscopy characterization, the wealth of the large number of samples and the conclusive discussion. A comprehensive picture emerges, based on a consistent experiment dataset, that provides further experimental support for the modern view on this catalyst. These results are important and I recommend acceptance of this nice piece of work.

However, before the paper is published, I would like to draw the author's attention to the following minor points:

1. The standard deviation of the activity measurements seems rather large (35%). Why?
2. The lines in Fig.4 can be debated. I recommend to change to dashed lines for the higher particle size end of the Cu and Cu/Zn-silicate curves (as done for the lower particle size end of the Cu/ZnO curve). Is it true that the Zn promotion seems to be the more important parameter compared to the structure sensitivity? I think this will be interesting to discuss for the colleagues doing catalyst synthesis. (This figure reminds me of the different catalyst "families" discussed by Muhler et al. in Catal. Lett. 2004, 92, 49.)
3. In the introduction it is stated correctly that there are numerous studies claiming structure and (Zn-)composition insensitivity (most of them quite old). It will be interesting for the reader to briefly mention in this context the new results of [43] and [Fichtl et al., Angew. Chem. 53 (2014) 7043] suggesting that the commonly used N₂O chemisorption method might not be a good way to determine the Cu surface area as it is likely not structure and (Zn-)composition insensitive by itself. Thus, the apparent discrepancy between the results reported here and these older reports might be resolved by the special characteristics of the copper surface determination method.
4. To contribute to the important discussion on the most suitable analysis methods for this catalyst: Is chemisorption data available for the catalysts reported here? Will there be a difference between TOFs calculated based on chemisorption and TEM size?
5. It is concluded that the Cu particles are monocrystalline due to the similar XRD and TEM sizes. Are HRTEM images available to exemplify this conclusion for each kind of catalyst (Cu, Cu/Zn-silicate and Cu/ZnO)?

Reviewer #3 (Remarks to the Author):

In this paper, the authors present an extensive comparison of the experimental activities of different size Cu and CuZn catalysts supported on various supports (SiO₂, C, and Al₂O₃) for methanol production. They conclude that when the size of the catalyst particles falls from 10 nm to 2 nm, the methanol production activity decreases by about 3 times, regardless of Zn loading or support. The authors suggest that this implies that a certain kind of site configuration, whose abundance decreases as particle size decreases, is the active site for the reaction.

Although it is good that many supports were tried, it appears that the authors did not attempt to address whether the very different supports utilized had differing influences on the activities of the catalysts. For smaller particles especially, the support may affect the electronic properties of the particles or the specific sites in close proximity between the metal and the support. The authors tend to assume (without rigorous rationalization) that the effect of support is similar regardless of its nature.

Additionally, it is noted that the smallest particles (2-5 nm), for which the authors noticed the decrease in activity, could only be synthesized on the SiO₂ support. It is thus even more important to disentangle the effect of the support with the effect of particle size in this case, as the conclusion hinges largely on the activity of these small particles. For example, looking at the case of Cu only catalysts (red line in figure 4), the activity is roughly constant going down from 15 to 5 nm, but steeply decreases for particles ~2 nm, where all samples in this range are supported on SiO₂. This decrease might thus be due to the combination of using a SiO₂ support and the small size of the particles, rather than just the latter alone.

For calculations of the TOF, the authors assumed that dispersion could be estimated by assuming fully accessible spherical particles. This could be a large source of error that would easily obscure the trends in TOF with size (as the activity is known to scale linearly with the number of accessible sites). For example, smaller particles might have a larger fraction of sites physically blocked due to binding to the support, and thus appear to have a smaller TOF. A more accurate measure of the dispersion would be welcome, especially given the main message of the paper.

Unfortunately, the manuscript reads more like an incoherent combination of arguments from the

literature with experimental data the authors gathered for this work. The arguments are far from convincing/leading to the conclusions the authors tend to derive. More specifically, and at places, conclusions appear to be self-contradictory: (e.g.: p. 11; lines 229-231 versus lines 249-251). All in all, the manuscript falls short of one's expectations for a journal such as Nature Communications.

Reviewer #4 (Remarks to the Author):

Technical report on the validity of the XRD data

This manuscript describes the structure-sensitivity of a large variety of copper based catalysts for the methanol synthesis reaction. The manuscript is well written and clearly reflects a meticulously performed study of a large and complex parameter set. In agreement with senior editor Dr. Enda Bergin, my comments concern the validity and conclusions based on the XRD data.

The XRD setup model, manufacturer, and radiation source are described in the manuscript. However, the goniometer type (theta-theta) and the applied optics are not. I would suggest to mention this in the supplementary. The employed size estimation method (Scherrer equation) is documented in the supplementary.

The XRD data are presented clearly in both manuscript and supplementary. It is not clear to me whether any background correction or smothering was applied to the presented data. The most important peaks (belonging to Cu, CuO, Cu₂O, Zn, and ZnO) are indicated on the figures. In case of Al₂O₃ as a support (figure 1 black and Supplementary figure 4 H) a peak is observed ~31deg. However, this is not indicated or commented in the manuscript or in the supplementary information.

Supplementary Table 1 summarizes the samples and main results. The manuscript describes that similar crystallite sizes were found for cobber oxide after heat treatment and for metallic copper after reduction and passivation and that a 15% size decrease is expected due to the difference in density of states for the two phases. However, this is not the case for Cu₁₀SG(3)-NO in Supplementary table 1, where the crystallite size of CuO is smaller (3.2nm) than the Cu (5.2nm). This is not discussed.

In a few cases (e.g. Cu₃S-NO) the crystallite sizes estimated by XRD are larger (10.6nm) than the particle sizes measured by (S)TEM (6.6nm). This cannot be explained by differences between crystallite size and particle size and is not commented.

Author response to the reviewers' comments:

Reviewer #1 (Remarks to the Author):

This is an interesting and very clearly written manuscript in which the authors endeavor to prove that the conversion of synthesis gas to methanol over copper catalysts is sensitive to the structure of the copper surface. An exhaustive study has been made of Cu particles of various sizes dispersed on zinc-free supports, and on zinc silicate and zinc oxide supports.

The main conclusions of the paper are:-

- (i) Zn acts as a promoter, and its efficacy depends on the thermodynamic stability of the zinc phase, with less stable zinc containing phases giving rise to bigger promotion effects.
- (ii) the turnover frequency for Cu containing particles decreases by a factor of ~3 on decreasing their mean size from 8 to 2nm, irrespective of the support identity.

Both of these trends are clearly demonstrated by the experimental results and subsequent analysis presented. In addition, the experimental techniques used are appropriate for the task in hand and are well executed. In fact, the meticulous and detailed presentation of the methods and results in the SI is to be commended.

We thank the reviewer for very favorable comments about our paper and the SI.

It would have been nice to include some higher (atomic) resolution imaging studies of the supported particles to confirm they were indeed Cu rather than CuOx in the STEM, and to investigate more definitively if their exposed surface facet planes, defect (e.g. twin) content, and substrate wetting characteristics were varying with particle size and support identity.

We have now added in situ HR-TEM images to the manuscript as well as to the SI (Figure 2 and Supplementary Figure 15). From these images and the derived lattice spacing it is evident that metallic copper prevails in these catalysts under reducing gas atmosphere. The in situ HR-TEM images show mono-crystalline copper particles. The copper particle sizes determined with TEM correspond to the crystallite sizes measured with XRD. This confirms the monocrystalline nature of the Cu particles. For the wetting characteristics of copper metal nanoparticles on silica we refer to our previous paper (van den Berg, R. *et al.* Support functionalization to retard Ostwald ripening in copper methanol synthesis catalysts. *ACS Catal.* **5**, 4439-4448 (2015)) in which it was demonstrated that the energy of the copper-silica interface does not depend on the gas environment, and that the contact angle does not depend on the size of the copper particles. For the wetting characteristics of Cu on zinc-containing supports we refer to the paper by Hansen et al. (Hansen, P. L. *et al.* Atom-resolved imaging of dynamic shape changes in supported copper nanocrystals. *Science* **295**, 2053-2055 (2002)), in which it was reported that the wetting characteristics of copper particles in the range from 3 to 6 nm are similar.

The changes to the manuscript are as follows:

Added experimental method for obtaining in situ HR-TEM images in the Supplementary information on page 15:

The images A and C in Figure 1, Figure 2 and Supplementary Figure 15 were acquired using an image-aberration corrected Titan 80-300 ETEM (FEI). Prior to an experiment, the image aberration corrector was tuned using a cross-grating (Agar S106) and the spherical aberration coefficient was set in the range of -10 to -20 μm . TEM samples of $\text{Cu}_2\text{S-N}_2$, $\text{Cu}_3\text{S-NO}$ and $\text{Cu}_{40}\text{SiO}_2$ were prepared by grinding and dispersing the resulting powder on stainless steel grids. The samples were inserted in the microscope using a Gatan heating holder (model 628). $\text{Cu}_2\text{S-N}_2$ was reduced at 350 $^\circ\text{C}$ at 1 mbar H_2 for 30 min and TEM images were subsequently acquired at these conditions with an electron dose-rate of 20 $\text{e}^-/(\text{\AA}^2\text{s})$ (Figure 1A). $\text{Cu}_3\text{S-NO}$ was reduced at 300 $^\circ\text{C}$ at 1 mbar H_2 for 30 min and TEM images were subsequently acquired at these conditions with an electron dose-rate of 100 $\text{e}^-/(\text{\AA}^2\text{s})$. Reduced and passivated $\text{Cu}_{40}\text{SiO}_2$ was re-reduced in the electron microscope at 250 $^\circ\text{C}$ at 1 mbar H_2 for 45 min and TEM images were subsequently acquired at these conditions with an electron dose-rate of 10 (Figure 1C) or 100 $\text{e}^-/(\text{\AA}^2\text{s})$ (Figure 2).

Added Figure 2 + caption:

Figure 2. High-resolution in situ TEM image of $\text{Cu}_{40}\text{SiO}_2$ at 1 mbar H_2 at 250 $^\circ\text{C}$. The scale bar corresponds to 5 nm. The image on the top right shows the Fast Fourier Transform of the TEM image. The imaged copper particle is mono-crystalline. The distance between the observed lattice fringes is 2.09 \AA , which corresponds to the (111) interplanar spacing of metallic copper crystals.

Added Supplementary Figure 15 + caption:

Supplementary Figure 15. High-resolution in situ TEM images of $\text{Cu}_3\text{S-NO}$ at 1 mbar H_2 at 300 $^\circ\text{C}$. The images on the right show the Fast Fourier Transforms corresponding to the TEM images on the left. The TEM images show that the copper particles are mono-crystalline. The distance between the observed lattice fringes is 2.09 \AA , which corresponds to the spacing of (111) planes in metallic copper crystals.

Added in the text at page 5:

High-resolution TEM images (Figure 2 and Supplementary Figure 15) of copper catalysts under a reducing gas atmosphere showed mono-crystalline particles with the lattice spacing of metallic copper.

Added in the text at page 9/10:

Neither for non-reducible oxide supports such as SiO_2 nor for carbon supports a strong interaction between metal particles larger than 2 nm and the support is expected.⁵⁰ This is confirmed by the large contact angle of 135° that has experimentally been found for copper particles on silica in reducing conditions.³⁴ Such a large contact angle allows approximating the copper surface area by assuming fully accessible spherical particles. Furthermore, it has been reported that the energy of the copper-silica interface does not depend on the gas environment, and that the contact angle of copper particles larger than 2 nm on silica does not depend on particle size.^{34,45} The observed particle size effect for the copper catalysts without zinc can thus not be due to support effects but rather seems to be due to changes in the structure of the copper surface.

For copper on zinc-containing supports, strong metal-support interactions have been reported in literature. It has been observed that for copper on ZnO both the shape of the copper particles as well as the coverage of the copper particles with (partly) reduced zinc species depend on the gas environment.^{27,45} The coverage with zinc species might depend on the copper particle size, and could result in a decreased activity for smaller copper particles.²⁷ Although we cannot fully exclude that support effects play a role in the observed particle size effect for zinc-containing supports, the similarity of the observed particle size effect for catalysts with and without zinc strongly suggests that the lower activity for small particles is mainly due to changes in the copper surface structure. Hence, decorating the copper surface with migrated (partially) reduced zinc species increases the activity up to a factor of 10, but does not seem to significantly alter the configuration of the active site.²⁸

Based on their experimental measurements of the dependence of turnover frequency on metal particle size, the authors postulate that "the methanol synthesis reaction predominantly takes place at surface sites with a unique configuration of several copper atoms such as step-edge sites, which smaller particles cannot accommodate." It would strengthen further the paper if some modelling studies more specific to this materials system and also incorporating the possible effects of Zn decoration of the Cu could be performed to complement this important postulate.

We agree with the reviewer that modelling studies are important to substantiate the claim that specific surface sites such as those at step edges are the active sites in methanol synthesis. We would like to point out that in references 17, 28, 39 and 53 extensive and relevant modelling work has already been reported. The main findings of these modelling studies are in detail discussed in relation to our experimental data on pages 11 and 12 and the possible effects of Zn decoration of the Cu is now discussed on pages 9 and 10.

Added in the text at page 9/10:

See added text in previous comment

The paper relates to an industrially important process which with a little more work could potentially be of interest to the readers of Nature Communications.

We thank Reviewer 1 for important and helpful comments to improve the manuscript.

Reviewer #2 (Remarks to the Author):

The paper by van den Berg et al. is a thorough and conclusive study of the structure and composition sensitivity of Cu(Zn) catalysts in the methanol synthesis reaction.

The authors present a large series of skillfully prepared and well characterized catalysts and report a clear trend of the TOF with absence or presence of the Zn promoter and - importantly - a breakdown in TOF for small Cu crystallites, similar to effects observed previously e.g. in ammonia synthesis over Ru.

Neither the effect of the Zn promoter nor the structure sensitivity by themselves are new, but the effect of very small particles is to my knowledge unprecedented for MeOH synthesis. I was impressed by the careful and detailed microscopy characterization, the wealth of the large number of samples and the conclusive discussion. A comprehensive picture emerges, based on a consistent experiment dataset, that provides further experimental support for the modern view on this catalyst. These results are important and I recommend acceptance of this nice piece of work.

However, before the paper is published, I would like to draw the author's attention to the following minor points:

1. The standard deviation of the activity measurements seems rather large (35%). Why?

We have collected the data of these almost 50 catalysts over a long period of time, with differences in catalyst amount, flows and conversion levels. Some measurements have been specifically repeated at rather low conversion levels (< 5%), in which case the inaccuracy was relatively large, also due to some varying adsorption of CO₂ by the metal carbonyl trap. Testing catalysts under identical conditions at conversion levels around 10% revealed a deviation of only 5-10%. Nevertheless we prefer to report the most conservative standard deviation of 35%. Since the observed effects in the paper are much larger than 35%, the uncertainty in the activity data does not influence our findings.

2. The lines in Fig.4 can be debated. I recommend to change to dashed lines for the higher particle size end of the Cu and Cu/Zn-silicate curves (as done for the lower particle size end of the Cu/ZnO curve).

The lines in now figure 5 serve as a guide to the eye for the observed trends. We agree with the reviewer that these lines can be debated. Based on the recommendation of the reviewer we have changed the lines to solid lines within the investigated size range and dashed lines outside of the investigated size range.

Changed Figure 5 and TOC.

Is it true that the Zn promotion seems to be the more important parameter compared to the structure sensitivity? I think this will be interesting to discuss for the colleagues doing catalyst synthesis. (This figure reminds me of the different catalyst "families" discussed by Muhler et al. in *Catal. Lett.* 2004, 92, 49.)

The reviewer points correctly out that the effects of Zn on the TOF are larger than that of size in the range studied. Control over Zn in the copper catalyst in that respect is more important than that of control over size. We note that the effect of Zn has been known for decades whereas the effects of size has been disputed and not settled. For the first time now the effects of size for both Cu and CuZn are reported under industrially relevant methanol synthesis conditions. Reference to the different catalyst 'families' discussed by Muhler in an earlier publication is made in the introduction (ref 24, *Catal. Lett.* **86**, 77-80 (2003)). In order to include the important comment of the reviewer we have changed the text in the manuscript as follows.

Added on page 9:

To obtain highly active copper-based methanol synthesis catalysts it is thus most important to promote catalysts with zinc as it can increase the activity by a factor of 10, but it is also important to tune the copper particle size as it can further increase the activity by a factor of 3.

3. In the introduction it is stated correctly that there are numerous studies claiming structure and (Zn-)composition insensitivity (most of them quite old). It will be interesting for the reader to briefly mention in this context the new results of [43] and [Fichtl et al., Angew. Chem. 53 (2014) 7043] suggesting that the commonly used N₂O chemisorption method might not be a good way to determine the Cu surface area as it is likely not structure and (Zn-)composition insensitive by itself. Thus, the apparent discrepancy between the results reported here and these older reports might be resolved by the special characteristics of the copper surface determination method.

We agree with the reviewer that the commonly used N₂O chemisorption method might comprise systematic errors. We now more clearly indicate in the manuscript that chemisorption is not the most reliable technique to establish Cu and CuZn particle sizes. We feel that this is particularly true for the smallest particle sizes (< 8 nm) in our work in view of fast equilibration of surface composition in response to changes in the thermodynamic potential of the gas phase.

Added in the text on page 2:

determined with N₂O Refractive Frontal Chromatography (N₂O-RFC)

Added in the text on page 3:

Recent studies have shown that N₂O is not just oxidizing the copper surface but also oxidizing (partly) reduced zinc species on metallic copper or oxygen vacancies in the zinc oxide support.²⁵⁻²⁷ The N₂O-RFC method therefore depends on the structure of the investigated catalyst and the conditions prior to N₂O exposure, and this might result in differences between the measured and the real exposed copper surface areas.

Added in the text on page 4:

As it has been shown that chemisorption techniques determining the copper surface area are sensitive to the catalyst structure and the pre-conditions, and in view of a presumably fast equilibration of the surface composition of especially the smallest particles (< 8 nm) in response to changes in the thermodynamic potential of the gas phase, we have chosen to investigate the structure sensitivity of the methanol synthesis reaction by relying on X-ray diffraction (XRD) and (scanning) transmission electron microscopy (S(TEM)) to determine the size and geometric surface area of the copper particles.

4. To contribute to the important discussion on the most suitable analysis methods for this catalyst:

Is chemisorption data available for the catalysts reported here? Will there be a difference between TOFs calculated based on chemisorption and TEM size?

For the reasons outlined above we have not extensively pursued chemisorption analysis. Furthermore the relatively low copper loadings in most of our samples enhance the statistical error of the chemisorption measurements. For that reason we have focused on TEM and XRD as more robust methods for determination of the copper particle size and specific copper surface areas of our catalysts.

5. It is concluded that the Cu particles are monocrystalline due to the similar XRD and TEM sizes. Are HRTEM images available to exemplify this conclusion for each kind of catalyst (Cu, Cu/Zn-silicate and Cu/ZnO)?

We have now added in situ HR-TEM images to the manuscript as well as to the SI (Figure 2 and Supplementary Figure 15). From these images and the derived lattice spacing it is evident that metallic copper prevails in these catalysts under reducing gas atmosphere. The in situ HR-TEM images show mono-crystalline copper particles. The copper particle sizes determined with TEM correspond to the crystallite sizes measured with XRD. This confirms the monocrystalline nature of the Cu particles.

The changes to the manuscript are as follows:

Added experimental method for obtaining in situ HR-TEM images in the Supplementary information on page 15:

The images A and C in Figure 1, Figure 2 and Supplementary Figure 15 were acquired using an image-aberration corrected Titan 80-300 ETEM (FEI). Prior to an experiment, the image aberration corrector was tuned using a cross-grating (Agar S106) and the spherical aberration coefficient was set in the range of -10 to -20 μm . TEM samples of $\text{Cu}_2\text{S-N}_2$, $\text{Cu}_3\text{S-NO}$ and $\text{Cu}_{40}\text{SiO}_2$ were prepared by grinding and dispersing the resulting powder on stainless steel grids. The samples were inserted in the microscope using a Gatan heating holder (model 628). $\text{Cu}_2\text{S-N}_2$ was reduced at 350 $^\circ\text{C}$ at 1 mbar H_2 for 30 min and TEM images were subsequently acquired at these conditions with an electron dose-rate of 20 $\text{e}^-/(\text{\AA}^2\text{s})$ (Figure 1A). $\text{Cu}_3\text{S-NO}$ was reduced at 300 $^\circ\text{C}$ at 1 mbar H_2 for 30 min and TEM images were subsequently acquired at these conditions with an electron dose-rate of 100 $\text{e}^-/(\text{\AA}^2\text{s})$. Reduced and passivated $\text{Cu}_{40}\text{SiO}_2$ was re-reduced in the electron microscope at 250 $^\circ\text{C}$ at 1 mbar H_2 for 45 min and TEM images were subsequently acquired at these conditions with an electron dose-rate of 10 (Figure 1C) or 100 $\text{e}^-/(\text{\AA}^2\text{s})$ (Figure 2).

Added Figure 2 and Figure caption:

Figure 2. High-resolution in situ TEM image of $\text{Cu}_{40}\text{SiO}_2$ at 1 mbar H_2 at 250 $^\circ\text{C}$. The scale bar corresponds to 5 nm. The image on the top right shows the Fast Fourier Transform of the TEM image. The imaged copper particle is mono-crystalline. The distance between the observed lattice fringes is 2.09 \AA , which corresponds to the (111) interplanar spacing of metallic copper crystals.

Added Supplementary Figure 15 and Figure caption:

Supplementary Figure 15. High-resolution in situ TEM images of Cu₃S-NO at 1 mbar H₂ at 300 °C.

The images on the right show the Fast Fourier Transforms corresponding to the TEM images on the left. The TEM images show that the copper particles are mono-crystalline. The distance between the observed lattice fringes is 2.09 Å, which corresponds to the spacing of (111) planes in metallic copper crystals.

Added in the text at page 5:

High-resolution TEM images (Figure 2 and Supplementary Figure 15) of copper catalysts under a reducing gas atmosphere showed mono-crystalline particles with a lattice spacing corresponding to metallic copper.

We thank Reviewer 2 for important and helpful comments to improve the manuscript.

Reviewer #3 (Remarks to the Author):

In this paper, the authors present an extensive comparison of the experimental activities of different size Cu and CuZn catalysts supported on various supports (SiO₂, C, and Al₂O₃) for methanol production. They conclude that when the size of the catalyst particles falls from 10 nm to 2 nm, the methanol production activity decreases by about 3 times, regardless of Zn loading or support. The authors suggest that this implies that a certain kind of site configuration, whose abundance decreases as particle size decreases, is the active site for the reaction.

Although it is good that many supports were tried, it appears that the authors did not attempt to address whether the very different supports utilized had differing influences on the activities of the catalysts. For smaller particles especially, the support may affect the electronic properties of the particles or the specific sites in close proximity between the metal and the support. The authors tend to assume (without rigorous rationalization) that the effect of support is similar regardless of its nature.

We thank the reviewer for pointing out that we should discuss more clearly the possible support effects in copper methanol synthesis catalysts. In general it has been found that for metal particles larger than 2 nm electronic effects of the support are limited. For reducible oxide supports phenomena such as strong metal support interaction (SMSI) may occur, but for non-reducible supports such as the silica and carbon supports used in this work an SMSI is not expected. We have now pointed this out in the main text with references to the relevant literature.

Additionally, it is noted that the smallest particles (2-5 nm), for which the authors noticed the decrease in activity, could only be synthesized on the SiO₂ support. It is thus even more important to disentangle the effect of the support with the effect of particle size in this case, as the conclusion hinges largely on the activity of these small particles. For example, looking at the case of Cu only catalysts (red line in figure 4), the activity is roughly constant going down from 15 to 5 nm, but steeply decreases for particles ~2 nm, where all samples in this range are supported on SiO₂. This

decrease might thus be due to the combination of using a SiO₂ support and the small size of the particles, rather than just the latter alone.

For calculations of the TOF, the authors assumed that dispersion could be estimated by assuming fully accessible spherical particles. This could be a large source of error that would easily obscure the trends in TOF with size (as the activity is known to scale linearly with the number of accessible sites). For example, smaller particles might have a larger fraction of sites physically blocked due to binding to the support, and thus appear to have a smaller TOF. A more accurate measure of the dispersion would be welcome, especially given the main message of the paper.

We agree with the reviewer that it is important to point out more clearly the accessibility of the copper surface of the nanoparticles supported on silica. In the past we have carried out contact angle measurements for reduced Cu/SiO₂ using E-TEM. From these measurements we have found that the contact angle is around 135 degrees showing that copper metal is only poorly wetting silica and virtually the entire copper nanoparticle is accessible for catalysis. Furthermore, it has been shown that the interfacial energy between copper and silica does not depend on the gas environment and that the contact angle of copper particles larger than 2 nm on silica does not depend on particle size (ACS Catal + Hansen). Next to that, no effect of support porosity or geometry on catalyst activity was observed (Supplementary Figure 18). This allows approximating the copper surface area by assuming fully accessible spherical particles. We have now discussed possible support effects in the text involving reference to this work using E-TEM.

Added in the text at page 9/10:

Neither for non-reducible oxide supports such as SiO₂ nor for carbon supports a strong interaction between metal particles larger than 2 nm and the support is expected.⁵⁰ This is confirmed by the large contact angle of 135° that has experimentally been found for copper particles on silica in reducing conditions.³⁴ Such a large contact angle allows approximating the copper surface area by assuming fully accessible spherical particles. Furthermore, it has been reported that the energy of the copper-silica interface does not depend on the gas environment, and that the contact angle of copper particles larger than 2 nm on silica does not depend on particle size.^{34,45} The observed particle size effect for the copper catalysts without zinc can thus not be due to support effects but rather seems to be due to changes in the structure of the copper surface.

For copper on zinc-containing supports, strong metal-support interactions have been reported in literature. It has been observed that for copper on ZnO both the shape of the copper particles as well as the coverage of the copper particles with (partly) reduced zinc species depend on the gas environment.^{27,45} The coverage with zinc species might depend on the copper particle size, and could result in a decreased activity for smaller copper particles.²⁷ Although we cannot fully exclude that support effects play a role in the observed particle size effect for zinc-containing supports, the similarity of the observed particle size effect for catalysts with and without zinc strongly suggests that the lower activity for small particles is mainly due to changes in the copper surface structure. Hence, decorating the copper surface with migrated (partially) reduced zinc species increases the activity up to a factor of 10, but does not seem to significantly alter the configuration of the active site.²⁸

Unfortunately, the manuscript reads more like an incoherent combination of arguments from the

literature with experimental data the authors gathered for this work. The arguments are far from convincing/leading to the conclusions the authors tend to derive. More specifically, and at places, conclusions appear to be self-contradictory: (e.g.: p. 11; lines 229-231 versus lines 249-251). All in all, the manuscript falls short of one's expectations for a journal such as Nature Communications.

We agree that the different possible factors that might have contributed to the observed particle size effects could have been more clearly explained. We have now included a discussion on the possible influence of the support and the presence of zinc on the copper particle size effect. We trust that the new text clearly delineates these effects and clarifies that based on the experimental results we have good arguments to state that most likely structural effects are dominant in the overall particle size effect of copper for the methanol synthesis under industrial conditions.

We thank Reviewer 3 for important and helpful comments to improve the manuscript.

Reviewer #4 (Remarks to the Author):

Technical report on the validity of the XRD data

This manuscript describes the structure-sensitivity of a large variety of copper based catalysts for the methanol synthesis reaction. The manuscript is well written and clearly reflects a meticulously performed study of a large and complex parameter set. In agreement with senior editor Dr. Enda Bergin, my comments concern the validity and conclusions based on the XRD data.

The XRD setup model, manufacturer, and radiation source are described in the manuscript. However, the goniometer type (theta-theta) and the applied optics are not. I would suggest to mention this in the supplementary. The employed size estimation method (Scherrer equation) is documented in the supplementary.

We have now added the goniometer type and the applied optics in the manuscript.

Revised XRD description at page 13/14 to:

X-ray diffraction was performed with a Bruker-AXS D2 Phaser powder X-ray diffractometer equipped with a Lynxeye detector, in Bragg-Brentano mode with a θ - θ system with a 141 mm radius. The radiation used is Co-K α 12 ($\lambda = 1.79026 \text{ \AA}$), operated at 30 kV, 10 mA.

The XRD data are presented clearly in both manuscript and supplementary. It is not clear to me whether any background correction or smothering was applied to the presented data. The most important peaks (belonging to Cu, CuO, Cu₂O, Zn, and ZnO) are indicated on the figures. In case of Al₂O₃ as a support (figure 1 black and Supplementary figure 4 H) a peak is observed $\sim 31^\circ$. However, this is not indicated or commented in the manuscript or in the supplementary information.

No background correction or smoothing was applied to the presented data. For Cu(Zn) on carbon or on silica the spectra have been normalized to the peak corresponding to the support. For Cu/ZnO(/Al₂O₃) the spectra are normalized on the highest peak. The peak around 31 degrees in Figure 4H corresponds to the lattice distance in carbon. For Cu/ZnO(/Al₂O₃) samples, carbon is used as a binder for pelletization. The information regarding the pelletization is now included in the supporting information and the origin of the peak around 31 degrees is given in the caption of what has now become Figure 3 and Supplementary Figure 4.

Revised caption Figure 3 at page 26:

The broad peak at $2\theta = 25^\circ$ for the CuZn/SiO₂ sample is due to the silica support and the peaks at 31° for the CuZn/C and CuZnO samples are due to the carbon support and the graphite lubricant used for pelletization, respectively.

Added at page 13 of the Supplementary Information:

No background correction or smoothing was applied.

Added at page 14 of the Supplementary Information:

For Cu(Zn) on carbon or on silica the spectra have been normalized to the peak corresponding to the support. For Cu/ZnO(/Al₂O₃) the spectra are normalized on the highest peak. The small peaks at 31° 2θ in image H are due to the graphite lubricant used for pelletization.

Revised at page 6 of the Supplementary Information:

The samples were extensively washed with water and dried at 80°C . The dried filter cake was crushed and sieved to 0.3 – 0.6 mm and calcined in a muffle oven at 325°C . The samples were subsequently pressed, pelletized, crushed and sieved to obtain the right sieve fraction for catalytic testing. For the pelletization of the samples 2 to 4% graphite was added as a lubricant.

Supplementary Table 1 summarizes the samples and main results. The manuscript describes that similar crystallite sizes were found for cobber oxide after heat treatment and for metallic copper after reduction and passivation and that a 15% size decrease is expected due to the difference in density of states for the two phases. However, this is not the case for Cu10SG(3)-NO in Supplementary table 1, where the crystallite size of CuO is smaller (3.2nm) than the Cu (5.2nm). This is not discussed.

Since the particles in the case of Cu10SG(3)-NO are below 6 nm, the long range crystallinity is limited. The intensity of the peaks corresponding to CuO for the calcined sample was therefore low (see Supplementary Figure 4A). For the reduced and passivated catalyst, the specimen holder was loaded in the glove box and subsequently sealed. The additional scattering due to the sealing cap reduced the relative intensity of the copper peaks further. The intensity of the peaks corresponding to Cu was therefore very low and only for the (111) diffraction just above the detection limit (see Supplementary Figure 4B). Since Cu particle sizes were estimated by applying the Scherrer equation

to the (111) diffraction, the accuracy of the metallic copper crystallite size of Cu₁₀SG(3)-NO is rather low, hence explaining the difference between the determined CuO and Cu crystallite size.

In a few cases (e.g. Cu₃S-NO) the crystallite sizes estimated by XRD are larger (10.6nm) than the particle sizes measured by (S)TEM (6.6nm). This cannot be explained by differences between crystallite size and particle size and is not commented.

XRD is a volume-averaged technique and is thus more sensitive to larger crystallites than to smaller crystallites. XRD therefore reports the volume-averaged crystallite size. For (S)TEM the number-averaged particle size is reported. In the case of samples with a relatively broad particle size distribution, e.g. Cu₃S-NO (see Supplementary Figure 5), the volume-averaged particle size can be significantly larger than the number-averaged particle size, while nevertheless corresponding to the same particle size distribution.

We thank Reviewer 4 for important and helpful comments to improve the manuscript with regard to the XRD data.

Added in the acknowledgement:

Dr. Jens Sehested is acknowledged for his suggestion to explore CuZn catalysts with different thermodynamic stabilities of the Zn precursors. Dr. Stig Helveg is thanked for his contribution to the electron microscopy studies.

REVIEWERS' COMMENTS:

Reviewer #1 (Remarks to the Author):

After carefully reading the revised paper, I am happy that the authors have satisfactorily addressed all the specific concerns that I raised in my initial review. They also seem to have made a decent job of answering the questions and criticisms posed by the other referees.

I believe the paper is now suitable for publication in Nature Communications.

Reviewer #2 (Remarks to the Author):

The authors have succeeded in further improving their paper during revision. All the points raised in my previous report have been addressed adequately and I think this also holds for the review reports of my referee colleagues. The addition TEM work is consistent with the main conclusions of this work and I recommend acceptance as is.